# Menopause and the Risk of Developing Age-Related Macular Degeneration in Korean Women

**DOI:** 10.3390/jcm11071899

**Published:** 2022-03-29

**Authors:** Jin-Sung Yuk, Je Hyung Hwang

**Affiliations:** 1Department of Obstetrics and Gynecology, Sanggye Paik Hospital, Inje University 1342, Dongil-ro, Nowon-gu, Seoul 139-707, Korea; cnnsbs@naver.com; 2Department of Ophthalmology, Sanggye Paik Hospital, Inje University 1342, Dongil-ro, Nowon-gu, Seoul 139-707, Korea

**Keywords:** age-related macular degeneration, cohort study, diabetes mellitus, menopause, risk factors

## Abstract

Previous studies have shown that menopausal hormone therapy in postmenopausal women results in a higher prevalence of age-related macular degeneration. This study aimed to evaluate the effects of menopause and patient factors on the development of age-related macular degeneration in Korean women. Data between 2011 and 2014 were collected from the Korean National Health Insurance database. In this retrospective cohort study, 97,651 participants were premenopausal and 33,598 were menopausal. Participants were divided into menopausal and premenopausal groups to analyze the risk factors associated with the development of age-related macular degeneration. The prevalence of age-related macular degeneration was compared between the two groups. Other patient factors were also analyzed. Using a 1:1 propensity score matching method and adjusting for variables, the incidence of age-related macular degeneration was not significantly different between the two groups. Age and diabetes mellitus were associated with an increased risk of developing age-related macular degeneration, regardless of menopause. Menopause was not a risk factor for age-related macular degeneration. These findings may help physicians identify women with diabetes who are at a greater risk of developing age-related macular degeneration.

## 1. Introduction

Age-related macular degeneration (AMD) is one of the major causes of severe and irreversible visual impairment in older patients [1,2]. The number of elderly patients with AMD and its prevalence are also increasing [3]. AMD is a multifactorial disease caused by environmental and genetic risk factors [4]. Previous studies have reported the following as risk factors for AMD: age, smoking, cardiovascular disease, diabetes mellitus (DM), drusen and pigment abnormalities, and variations in the complement factors B and H [5,6,7,8]. Estrogen has also been revealed to be a risk factor for AMD in previous epidemiologic studies [9,10]. In contrast, the Eye Disease Case-Control Study Group suggested that greater estrogen exposure may be associated with a lower prevalence of AMD [11]. Rudnicka et al. suggested that differences in estrogen levels between men and women could be the cause of gender differences in the incidence of AMD [10]. Estrogens are mediated by two estrogen receptors (α and β); estrogen receptor α is present in the retina of females [12]. Retinal pigment epithelium (RPE) is a critical tissue in AMD aggravation [13]. Genes regulated by estrogen-mediated mechanisms are expressed in the RPE to participate in extracellular matrix turnover. Such regulation may contribute to finding the pathogenic link between estrogen status and AMD incidence [14]. Menopause is the permanent cessation of menstruation, resulting in the loss of ovarian follicle development [15]. Most menopausal women experience vasomotor symptoms, including hot flushes, chills, anxiety, sleep disruption, and palpitations. Although the pathophysiology of vasomotor symptoms remains poorly characterized, these symptoms are known to be caused by estrogen withdrawal [16]. Early menopause is associated with AMD development [17,18], implying that estrogen withdrawal may be related to the development of AMD [19]. However, there is a limited understanding of the effect of estrogens on AMD development. Moreover, the results from epidemiological studies showing the relationship between human female hormones and the risk of developing AMD are limited and inconsistent.

The aim of this study aimed was to assess the relationship between menopause and AMD prevalence in Korean women.

## 2. Materials and Methods

### 2.1. Database

All Koreans are obligated by law to register in national health insurance. The National Health Insurance Service (NHIS) stores medical information (age, sex, diagnosis, drug prescription, surgery types, types of medical insurance, hospitalization, and outpatient examinations) of subscribed Koreans (about 51 million). The Health Insurance Review and Assessment Service (HIRA) is an institution that examines the adequacy of medical expenses to prevent disputes over insurance payments between NHIS and hospitals. The HIRA shares most of the medical record information stored by the NHIS. This population-based retrospective cohort study used the medical data of HIRA gathered between 1 January 2007 and 31 December 2020. The data of 154,781 patients who underwent national examinations by the Korean Health Insurance Corporation from 2011 to 2014 were extracted.

### 2.2. Selection of Participants

The International Classification of Diseases, 10th revision (ICD-10), and Korea Health Insurance Medical Care Expenses (2016, 2019 version) were used in this study to determine adequate participants and outcome variables. The subjects of this study were women aged between 40 and 59 years who visited medical institutions for medical checkups between 1 January 2011 and 31 December 2014. Among these women, those who visited medical institutions with menopause-related diagnostic codes (N95.x, M80.0, M81.0, E28.3) more than twice were defined as the menopausal group. To increase accuracy, subjects who visited hospitals and who were assigned diagnostic codes related to menopause in two (or more) different occasions were defined as menopause groups. In addition, the date on which subjects were first assigned a menopause diagnosis code was considered as the starting date of menopause. The remaining subjects were included in the control group. Subjects with previous diagnostic codes of cancer (any Cxx) or macular diseases (H30-36) were excluded based on the first post-examination duration of 180 days in both groups. Among the selected menopausal and control groups, a 1:1 propensity score matching was conducted for multiple variables, such as the year of inclusion, age, socioeconomic status, region, Charlson Comorbidity Index, parity, cardiovascular disease, hypertension, diabetes, dyslipidemia, and antithrombotic agent use.

### 2.3. Outcome

AMD was defined in participants who visited the medical institutions with the AMD diagnostic code (H35.31) more than three times. The H35.31 diagnostic code can be used after AMD is confirmed through ophthalmologists’ examinations, which include fundus photography, optical coherence tomography, and fluorescein fundus angiography.

### 2.4. Variables

Subjects were defined as having a low socioeconomic status when the medical insurance type was medical aid. Rural area subjects were defined as those from non-metropolitan regions. The Charlson Comorbidity Index was obtained using diagnostic codes from the first examination date to one year prior. Parity was defined as childbirths within the cohort. Cardiovascular disease (I21–23, I60–64, I11.0, I13.0, I13.2, I25.5, I42, I50), hypertension (I10–15), diabetes (E10–14), dyslipidemia (E78), and menopause (N95, N80.0, M81.0, E28.3) were defined as those who visited the medical institutions at least twice with the corresponding diagnostic codes. Antithrombotic agent use before inclusion were defined as those who used antithrombotic agents before the first examination date (unfractionated heparin, low-molecular-weight heparin, fondaparinux, warfarin, aspirin, direct oral anticoagulants, clopidogrel, and cilostazol) and took them for more than 180 days.

### 2.5. Statistical Analysis

All statistical analyses were performed using SAS Enterprise Guide 7.15 (SAS Institute Inc, Cary, NC, USA) and R 3.5.1 (The Foundation for Statistical Computing, Vienna, Austria). All analyses were considered statistically significant when *p* = 0.05. Two tests were also performed: the Cochran-Mantel-Haenszel test for the analysis of categorical variables and the Wilcoxon signed-rank test for the analysis of continuous variables. Standardized differences were used to evaluate the variables subjected to matching. Conditional logistic regression analysis was performed to determine the contribution of menopause to the risk of developing AMD and subsequently adjusted for confounding factors. The listwise deletion method was used to process missing values. Conditional logistic regression analysis was conducted for those in rural areas to calculate the risk of developing AMD to confirm the robustness of our research.

### 2.6. Ethics

This study was approved by the Institutional Review Board of the Sanggye Paik Hospital (approval number 202202003). The HIRA removes individual identifying variables before providing medical information to researchers (de-identification). Furthermore, the HIRA authorizes analyses only on closed servers and thereafter allows the resultant tables, figures, and numbers to be extracted from the server. Therefore, the anonymity of the study participants was preserved. Moreover, informed consent was not required in accordance with the Bioethics and Safety Act of South Korea. Following HIRA’s personal information protection policy, only research results can be taken out of the server; therefore, the raw data cannot be provided to the readers.

The data used in this study were provided by HIRA. However, we declare that HIRA has no conflicts of interest in this study.

## 3. Results

### 3.1. Patient Demographics

The data of 154,781 patients who had undergone national examinations by the Korean Health Insurance Corporation from 2011 to 2014 were extracted. Among these patients, 97,651 were classified into the premenopausal group, and 33,598 patients were classified into the menopausal group (Figure 1). The median age of these patients was 47 (range: 43–52) years in the premenopausal group and 54 (range: 51–56) years in the menopausal group (Appendix A). After 1:1 propensity score matching, 32,887 participants were included in the premenopausal and menopausal groups, respectively (Table 1). The average age after propensity score matching was 54 (range: 51–57) years and 54 (range: 51–56) years in the premenopausal and menopausal groups, respectively. The additional features of the participants are presented in Table 1.

### 3.2. Menopause and AMD

The number of patients with AMD was 51 (0.2%) and 50 (0.2%) in the premenopausal and menopausal groups, respectively; AMD incidence in both groups was not different (Table 2). The odds ratio was 0.98 (0.664–1.448, *p* = 0.921) before the adjustment of the variables. After adjusting for age per 5 years, socioeconomic status, region, Charlson comorbidity index, and parity, the odds ratio was 1.00 (0.67–1.492, *p* = 1), and after additional adjustment for cardiovascular disease, hypertension, diabetes mellitus, dyslipidemia, and antithrombotic agent use, the odds ratio was 1.011 (0.671–1.524, *p* = 0.959) (Figure 2, Table 3). Kaplan–Meier estimation showed no difference in the event-free survival rates between the two groups (Appendix A).

Formula (1): ORs adjusted for age per 5 years, SES, region, CCI, and parity.

Formula (2): ORs adjusted for age per 5 years, SES, region, CCI, parity, CVD, hypertension, DM, dyslipidemia, and antithrombotic agent after inclusion.

CCI, Charlson comorbidity index; CI, confidence interval; CVD, cardiovascular disease; DM, diabetes mellitus; MHT, menopausal hormone therapy; OR, odds ratio; SES, socioeconomic status

### 3.3. Other Risk Factors and AMD

Odds ratio analysis after adjusting for age per 5 years, socioeconomic status, region, Charlson comorbidity index, parity, cardiovascular disease, hypertension, diabetes mellitus, dyslipidemia, and antithrombotic agent use showed that aging increased the risk of developing AMD every 5 years. Meanwhile, the odds ratio for diabetes mellitus was 1.638 (1.027–2.614, *p* > 0.05) (Appendix A).

## 4. Discussion

In this study involving large-scale Korean Health Insurance Corporation data of Korean women, the association of various risk factors—including menopause—with the development of AMD was analyzed. After adjusting for multiple variables (age per 5 years, socioeconomic status, region, Charlson comorbidity index, parity, cardiovascular disease, hypertension, diabetes mellitus, dyslipidemia, and antithrombotic agent use), menopause was not associated with the risk of developing AMD. Conversely, older age and diabetes mellitus increased the risk of developing AMD.

Estrogen is one of the most important reproductive hormones in women, and estrogen and progesterone receptor mRNAs are present in the human retina [20]. However, the relationship between estrogen receptors and AMD development is currently unknown. Previous epidemiological studies have shown that the relationship between estrogen exposure and AMD development is inconsistent. The Eye Disease Case-Control Group showed that menopausal hormone therapy decreased the risk of developing AMD [11]. Also, Patnaik et al. reported that menopausal hormone therapy plays a beneficial role in preventing AMD [21]. Furthermore, Tomany et al. reported an increased development of AMD in patients with early menopause following oophorectomy [22]. Moreover, other population-based studies, such as the Aravind Comprehensive Eye Survey and the Blue Mountain Eye Study, showed that a shorter duration of estrogen exposure may increase the risk of developing AMD [19,23]. These findings support the hypothesis that a longer duration of estrogen exposure may decrease the risk of developing AMD.

In contrast, no similar relationship was observed in this study. The Beaver Dam Eye Study and the Salisbury Eye Evaluation Project reported similar results to our study; they reported no association between estrogen and AMD development [17,24,25]. Similarly, Feskanich et al. reported that endogenous estrogen exposure was not associated with AMD development [26]. However, each population-based cross-sectional study was limited by uncontrolled confounding factors and selection biases because the data were from single hospitals with relatively small subject numbers.

The AMD incidence was relatively low in this study. This is due to the younger average age of the participants, compared to those reported in other studies (neovascular AMD: 0.6%, 0.1% geographic atrophy in Cho et al. [27] and 2.6% in the Salisbury Eye Evaluation Project [25]). Furthermore, since all patients aged over 60 years were menopausal, it was impossible to verify whether menopause was a significant risk factor for AMD development within similar age groups.

As was already known, age was the strongest risk factor for AMD development. In this study, the odds ratio for developing AMD regardless of menopause was 5.95 (1.617–21.891) in patients aged 45–49 years, 16.218 (4.636–56.728) in patients aged 50–54 years, and 28.412 (8.123–99.378) in patients aged 55–59 years.

Diabetes is a risk factor for AMD, and previous studies have shown that diabetes increases the risk of developing AMD [28,29]. Conversely, Clemons et al. [30] reported that diabetes plays a protective role in AMD. In this study, diabetes increased the risk of AMD development (odds ratio 1.638 [1.027–2.614], *p* = 0.038) regardless of menopause, after adjusting for multiple variables. In addition, socioeconomic status, region, Charlson comorbidity index, parity, cardiovascular disease, hypertension, dyslipidemia, and antithrombotic agent use were not associated with AMD development.

The advantages of this study compared with previous studies are as follows: First, we performed 1:1 propensity score matching. Second, this was one of the largest-scale Asian population studies. Third, our data were acquired from the National Health Screening Program and Insurance data provided by the NHIS; thus, this study included participants from all regions of Korea. Fourth, we adjusted for various variables (age per 5 years, socioeconomic status, region, Charlson comorbidity index, parity, cardiovascular disease, hypertension, diabetes mellitus, dyslipidemia, and antithrombotic agent use).

This study had several limitations. First, considering that macular degeneration is most common in the elderly population, participants included in this study were relatively young (40–59 years). Subjects included in this study were patients diagnosed with menopause between 1 January 2011 and 31 December 2014. However, observation was conducted until 31 December 2020, resulting in a follow-up period of up to 10 years. Therefore, while the median age when selecting both groups was 54 years old, it reached 60 years old (or older) in 2020. Ultimately, the bias associated with age was reduced through an extended observation period. Second, the AMD types could not be identified because the AMD diagnostic codes used in this study did not differentiate between neovascular AMD and geographic atrophy. Third, the severity of AMD could not be assessed. Fourth, the study data had a selection bias for analyzing risk factors for AMD, except for menopause, because it included only female patients. Fifth, the effects of menopausal hormone treatments were not analyzed. Finally, we were unable to consider risk factors for AMD—such as smoking, nutrition, and genetic risk factors—in our analysis due to the unavailability of such data. Therefore, further research is needed to address this problem.

In conclusion, menopause is not associated with AMD development. However, aging and diabetes mellitus are risk factors for developing AMD. Thus, these findings may help physicians identify women with diabetes who are at a greater risk of developing AMD. This study analyzed the data of 150,000 participants from the Korean Health Insurance Corporation, including 130,000 subjects addressing the risk factors of AMD. This was one of the largest-scale studies in the relevant research and may be beneficial for understanding the effect of menopause on AMD development.

## Figures and Tables

**Figure 1 jcm-11-01899-f001:**
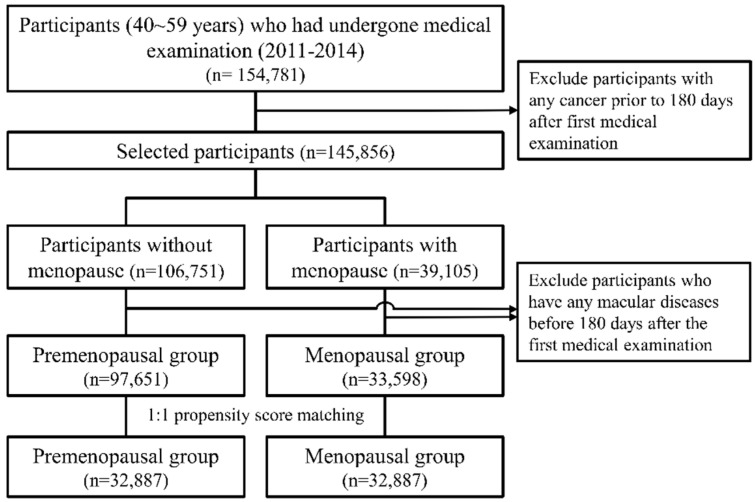
Flowchart for the selection of participants from the Korea National Health Insurance Database, 2007–2020.

**Figure 2 jcm-11-01899-f002:**
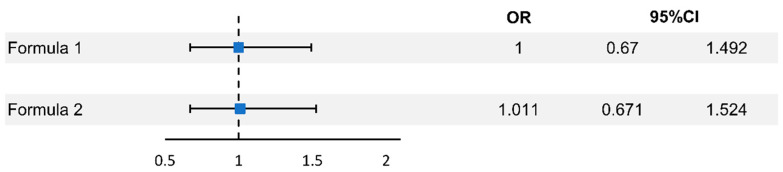
Odds ratios for risk of age-related macular degeneration according to menopause. Korea National Health Insurance Database, 2007–2020.

**Table 1 jcm-11-01899-t001:** Characteristics of participants according to whether or not menopausal status was performed after matching. Korea National Health Insurance Database, 2007–2020.

	Non-Menopause	Menopause	Total	*p*-Value	Standardized Difference
Nunber of women	32,887	32,887	65,774		
Median age (years)	54 (51–57)	54 (51–56)	54 (51–56)	0.2	−0.024
Age at inclusion (years)				0.031	0.033
40–44	1111 (3.4)	1087 (3.3)	2198 (3.3)		
45–49	4661 (14.2)	4675 (14.2)	9336 (14.2)		
50–54	12,962 (39.4)	13,452 (40.9)	26,414 (40.2)		
55–59	14,153 (43)	13,673 (41.6)	27,826 (42.3)		
Year at inclusion				<0.001	0.033
2011	7421 (22.6)	6991 (21.3)	14,412 (21.9)		
2012	7857 (23.9)	7894 (24)	15,751 (23.9)		
2013	8912 (27.1)	9150 (27.8)	18,062 (27.5)		
2014	8697 (26.4)	8852 (26.9)	17,549 (26.7)		
SES				0.724	−0.003
Mid~high SES	31,952 (97.2)	31,967 (97.2)	63,919 (97.2)		
Low SES	935 (2.8)	920 (2.8)	1855 (2.8)		
Region				<0.001	−0.036
Urban area	17,967 (54.6)	18,555 (56.4)	36,522 (55.5)		
Rural area	14,920 (45.4)	14,332 (43.6)	29,252 (44.5)		
CCI				<0.001	0.038
0	21,496 (65.4)	20,895 (63.5)	42,391 (64.4)		
1	6038 (18.4)	6373 (19.4)	12,411 (18.9)		
≥2	5353 (16.3)	5619 (17.1)	10,972 (16.7)		
Parity in cohort				0.142	0.012
0	32,820 (99.8)	32,801 (99.7)	65,621 (99.8)		
1	47 (0.1)	60 (0.2)	107 (0.2)		
≥2	20 (0.1)	26 (0.1)	46 (0.1)		
Cardiovascular disease before inclusion				0.804	0.002
Absent	31,452 (95.6)	31,439 (95.6)	62,891 (95.6)		
Present	1435 (4.4)	1448 (4.4)	2883 (4.4)		
Hypertension before inclusion				0.045	−0.016
Absent	23,752 (72.2)	23,981 (72.9)	47,733 (72.6)		
Present	9135 (27.8)	8906 (27.1)	18,041 (27.4)		
DM before inclusion				0.009	0.023
Absent	28,497 (86.7)	28,267 (86)	56,764 (86.3)		
Present	4390 (13.3)	4620 (14)	9010 (13.7)		
Dyslipidemia before inclusion				<0.001	0.062
Absent	20,607 (62.7)	19,616 (59.6)	40,223 (61.2)		
Present	12,280 (37.3)	13,271 (40.4)	25,551 (38.8)		
First antithrombotic agent before inclusion				0.284	0.008
Absent	30,763 (93.5)	30,695 (93.3)	61,458 (93.4)		
Present	2124 (6.5)	2192 (6.7)	4316 (6.6)		

DM, diabetes mellitus; CCI, Charlson comorbidity index; SES, socioeconomic status.

**Table 2 jcm-11-01899-t002:** The incidence of age-related macular degeneration with/without menopause. Korea National Health Insurance Database, 2007–2020.

	Pre-Menopause	Menopause	Total	*p*-Value
Number of women	32,887	32,887	65,774	
Age-related macular degeneration				0.889
Absent	32,836 (99.8)	32,837 (99.8)	65,673 (99.8)	
Present	51 (0.2)	50 (0.2)	101 (0.2)	

**Table 3 jcm-11-01899-t003:** Odds ratios for risk of age-related macular degeneration according to menopause after matching. Korea National Health Insurance Database, 2007–2020.

	Unadjusted		Formula (1) ^a^		Formula (2) ^b^	
	OR (95% CI) ^a^	*p*-value	OR (95% CI) ^a^	*p*-value	OR (95% CI) ^a^	*p*-value
Menopause	0.98 (0.664–1.448)	0.921	1 (0.67–1.492)	1	1.011 (0.671–1.524)	0.959

CCI, Charlson comorbidity index; CI, confidence interval; CVD, cardiovascular disease; DM, diabetes mellitus; MHT, menopausal hormone therapy; OR, odds ratio; SES, socioeconomic status. ^a^ ORs were adjusted for age per 5 years, SES, region, CCI, parity. ^b^ ORs were adjusted for age per 5 years, SES, region, CCI, parity, CVD, hypertension, DM, dyslipidemia, use of an antithrombotic agent.

## Data Availability

All data generated or analyzed during this study are included in this article. Further enquiries can be directed to the corresponding author.

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
