# Peer review of "Menopause and the Risk of Developing Age-Related Macular Degeneration in Korean Women"

_jcm, 2022, doi:10.3390/jcm11071899_

Round 1

Reviewer 1 Report

The manuscript has been improved

Author Response

Reviewer 1 report:

The manuscript has been improved

We would like to thank the reviewer for their positive feedback on our manuscript. We have carefully revised our manuscript to address all previous comments and are glad to note that the quality of our revised manuscript is satisfactory. We can confirm that the revised manuscript has now been edited once more by a native English speaker at Editage to maintain manuscript quality following the changes made to the text. Please find attached the proofreading certification.

Reviewer 2 Report

Clinically, there are gender differences in the incidence of AMD. Hormone therapy was considered to reduce the risk of advanced development of macular degeneration within the Women population. In addition, artificial "early" menopause was found to increase the risk of macular degeneration. Despite these sporadic studies, the association between menopause and the incidence of AMD remains largely uncertain. In the present clinical study, 154,781 patients were selected and divided into the pre-menopausal and menopausal groups. There are statistical differences in AMD incidence found between the two groups as in prior studies. Overall, the study is unsurprised but clearly presented.

Author Response

Reviewer 2 report:

Clinically, there are gender differences in the incidence of AMD. Hormone therapy was considered to reduce the risk of advanced development of macular degeneration within the Women population. In addition, artificial "early" menopause was found to increase the risk of macular degeneration. Despite these sporadic studies, the association between menopause and the incidence of AMD remains largely uncertain. In the present clinical study, 154,781 patients were selected and divided into the pre-menopausal and menopausal groups. There are statistical differences in AMD incidence found between the two groups as in prior studies. Overall, the study is unsurprised but clearly presented.

We would like to thank the reviewer for their positive feedback on our manuscript. We have carefully revised our manuscript to address all previous comments and are glad to note that the quality of our revised manuscript is satisfactory. We can confirm that the revised manuscript has now been edited once more by a native English speaker at Editage to maintain manuscript quality following the changes made to the text. Please find attached the proofreading certification.

Reviewer 3 Report

The paper “Menopause and the risk of developing age related macular degeneration in Korean women” aimed to assess the role of menopause as risk factor for AMD. This study is based on data  collected in the NHS in almost 51 millions Koreans and the present study collected data from more than 150 000 individuals who underwent national examination. Authors concluded that menopause does not play a role as risk factors for AMD

There are major issues in the manuscript mainly about the definition of menopause for the population and the selected population to perform the analysis.

  • It is not clear how the diagnosis of menopause was made and the duration of the menopause. We can guess that rather the status of menopause, it is the hormonal status and the duration of this hormonal status that should play a role for AMD. It is unlikely that menopause plays a role as on/off factors (ie present or not) for AMD that is a long lasting disease. If menopause is self-declared, it needs to be mentioned.
  • The role of hormonal status is mentioned but there is no information about hormonal substitution. We do not know how many patients were treated for their menopause. Some women are probably treated for their trouble linked to menopause but there are included in the same group as those who are not treated.
  • The diagnosis code H35.31 seems to correspond to early forms of AMD. How this diagnosis was made (through color photographs, from self-declared diagnosis). It is likely that early stages as underdiagnosed.
  • There is a huge bias with the comparative group: as people from menopause group are older than non menopause group, I agree that multivariate analysis can take it into account for analysis. But to obtain matched groups, group from menopause group included mostly the youngest while for non menopause group, included the oldest. So the groups were matched but people from menopause group had probably a very short duration with hormonal changes and as mentioned in point 1) it can play a role.
  • AMD is a multifactorial disease and it is very questionable not to consider other risk factors like smoking, nutrition and genetic risk factors. It is due to the design and the data collected in medical record information but when considering only 32 000 women, it is really questionable.
  • Age of women included in the study is definitely too low. This is due to the non menopause group that needs to be matched to the menopause group. However, it is not representative of AMD population with almost 60% < 55y old women

Author Response

Reviewer 3 report:

The paper “Menopause and the risk of developing age related macular degeneration in Korean women” aimed to assess the role of menopause as risk factor for AMD. This study is based on data collected in the NHS in almost 51 millions Koreans and the present study collected data from more than 150 000 individuals who underwent national examination. Authors concluded that menopause does not play a role as risk factors for AMD

We would like to thank the reviewer for their valuable comments regarding our manuscript, which we believe have helped us improve our manuscript and provide a more balanced account of our research. We have carefully revised our manuscript according to the reviewer’s suggestions and made the necessary changes, which have been highlighted in yellow in the revised manuscript to facilitate the review process. Please find below our point-by-point responses to all comments.

There are major issues in the manuscript mainly about the definition of menopause for the population and the selected population to perform the analysis.

It is not clear how the diagnosis of menopause was made and the duration of the menopause. We can guess that rather the status of menopause, it is the hormonal status and the duration of this hormonal status that should play a role for AMD. It is unlikely that menopause plays a role as on/off factors (ie present or not) for AMD that is a long lasting disease. If menopause is self-declared, it needs to be mentioned.

We would like to thank the reviewer for their valuable comment and apologize for the lack of clarity. In this study, menopause was determined when the associated diagnostic code was provided by the doctor. To increase accuracy, subjects who visited hospitals and who were assigned diagnostic codes related to menopause in two (or more) different occasions were defined as menopause groups. In addition, the date on which subjects were first assigned a menopause diagnostic code was considered as the starting date of menopause. To increase clarity and address your comment, we have now added this content to our revised manuscript (line 90~94).

The role of hormonal status is mentioned but there is no information about hormonal substitution. We do not know how many patients were treated for their menopause. Some women are probably treated for their trouble linked to menopause but there are included in the same group as those who are not treated.

We would like to thank the reviewer for their insightful comment and apologize for the lack of information. As the reviewer correctly pointed out, the effects of menopausal hormone treatments should be considered. However, research is warranted to first determine whether menopause is a risk factor for age-related macular degeneration (AMD). Therefore, we believe that the influence of menopausal hormone treatments should be investigated as a follow-up study of the present research. Nevertheless, this has now been added as a limitation to our revised manuscript (line 303~304) .

The diagnosis code H35.31 seems to correspond to early forms of AMD. How this diagnosis was made (through color photographs, from self-declared diagnosis). It is likely that early stages as underdiagnosed.

We would like to thank the reviewer for their valuable comment. As AMD is considered as an incurable disease, medical insurance in South Korea supports a large portion of the patients' medical expenses, including drug costs. The H35.31 diagnostic code can be used after AMD is confirmed through ophthalmologists’ examinations, which include fundus photography, OCT, and FAG. However, this diagnostic code does not differentiate between dry and wet AMD. Furthermore, it does not allow researchers to verify whether the injection treatment was conducted. These limitations have now been added to our revised manuscript to increased accuracy and clarity. Nevertheless, AMD was confirmed in patients included in this study through fundus photographs, OCT, and FAG examinations conducted by expert retinal specialists (line 107~110).

There is a huge bias with the comparative group: as people from menopause group are older than non menopause group, I agree that multivariate analysis can take it into account for analysis. But to obtain matched groups, group from menopause group included mostly the youngest while for non menopause group, included the oldest. So the groups were matched but people from menopause group had probably a very short duration with hormonal changes and as mentioned in point 1) it can play a role.

We would like to thank the reviewer for their valuable comment. Subjects included in this study were patients diagnosed with menopause between January 1, 2011 and December 31, 2014. However, observation was conducted until December 31, 2020, resulting in a follow-up period of up to 10 years. Therefore, while the median age in this study, when selecting both groups, was 54 years old (Table 1), it reached 60 years old (or older) in 2020. Ultimately, the bias associated with age was reduced through an extended observation period.(line 289~298)

AMD is a multifactorial disease and it is very questionable not to consider other risk factors like smoking, nutrition and genetic risk factors. It is due to the design and the data collected in medical record information but when considering only 32 000 women, it is really questionable.

We would like to thank the reviewer for their valuable comment. This study is based on health insurance data, which does not contain information related to smoking, nutrition, and genetic risk factors. Therefore, we are unfortunately unable to take into consideration these factors when conducting our analysis. Nevertheless, this limitation has now been added to our revised manuscript to increase accuracy (line 304~306).

Age of women included in the study is definitely too low. This is due to the non menopause group that needs to be matched to the menopause group. However, it is not representative of AMD population with almost 60% < 55y old women

We would like to thank the reviewer for their valuable comment. We targeted patients aged ~50 years old (i.e., the average age of menopause onset) in order to determine the risk factors for menopause. If women in their 60s had been included instead, most of the subjects would have already been diagnosed with menopause. Considering that macular degeneration is most common in the elderly population, the point the reviewer raised has now been added to our revised manuscript as a limitation of the present study. As mentioned earlier, subjects included in this study were patients diagnosed with menopause between January 1, 2011 and December 31, 2014. However, observation was conducted until December 31, 2020, resulting in a follow-up period of up to 10 years. Therefore, while the median age in this study, when selecting both groups, was 54 years old (Table 1), it reached 60 years old (or older) in 2020. Ultimately, the bias associated with age was reduced through an extended observation period (line 289~298).

Round 2

Reviewer 3 Report

The answers clarified the manuscript. There are bisas but at least it is mentioned more clearly.